# Event-Guided Rolling Shutter Correction with Time-Aware Cross-Attentions

## ABSTRACT

Many consumer cameras with rolling shutter (RS) CMOS would suffer undesired distortion and artifacts, particularly when objects experiences fast motion. The neuromorphic event camera, with high temporal resolution events, could bring much benefit to the RS correction process. In this work, we explore the characteristics of RS images and event data for the design of the rolling shutter correction (RSC) model. Specifically, the relationship between RS images and event data is modeled by incorporating time encoding to the computation of cross-attention in transformer encoder to achieve time-aware multi-modal information fusion. Features from RS images enhanced by event data are adopted as keys and values in transformer decoder, providing source for appearance, while features from event data enhanced by RS images are adopted as queries, providing spatial transition information. By embedding the time information of the desired GS image into the query, the transformer with deformable attention is capable of producing the target GS image. To enhance the model's generalization ability, we propose to further self-supervise the model by cycling between time coordinate systems corresponding to RS images and GS images. Extensive evaluations over both synthetic and real datasets demonstrate that the proposed method performs favorably against state-of-the-art approaches.

## CCS CONCEPTS

• **Computing methodologies** → **Computer vision**; **Reconstruction**.

## KEYWORDS

Event Camera, Multi-Modality, Image Enhancement, Rolling Shutter Correction, Self-supervised Training

## 1 INTRODUCTION

Many consumer cameras, such as smartphone cameras, employ a rolling shutter (RS) CMOS sensor for power and cost consideration. The rolling shutter camera sensor exposes asynchronously in a row-by-row manner while the global shutter exposes at all positions simultaneously. Hence the images based on RS exposure would suffer image distortion and occlusion problems, which is called jelly effect, particularly when objects experience fast motion. To remove such artifacts in an image caused by rolling shutter and recover a

*ACM MM, 2024, Melbourne, Australia*

© 2024 Copyright held by the owner/author(s). Publication rights licensed to ACM.
ACM ISBN 978-x-xxxx-xxxx-x/YY/MM
https://doi.org/10.1145/nnnnnnn.nnnnnnn

distortion-free GS image, various rolling shutter correction (RSC) techniques are developed.

The task of rolling shutter correction is challenging because it is difficult to estimate the transformation between RS and GS images especially when rolling shutter is coupled with both object motion and camera motion. It is a highly ill-posed problem for RSC from a single image due to the absence of motion information. Hence, there have been several works [5, 15, 31] proposed to first estimate motion fields between consecutive frames and compute correction fields based on them, which is followed by applying warping operation to the RS images for GS image recovery.

However, most of them exhibit poor performance when confronted with complex motion because of simple motion assumption during exposure.

Event camera, a bio-inspired sensor that outputs event signal with high temporal resolution (in the order of μs) by asynchronously measuring brightness change at each pixel [6], could provide much complementary information for the RSC task. By synergizing a RGB camera and an event camera, the triggered events are able to record how intensity at each pixel evolves during exposure of RGB camera. They carry abundant information about motion happened in the scene. Hence, making full use of events could bring much benefit to the process of rolling shutter correction. There have been a few attempts along this direction. Zhou *et al.* [32] present a two-branch approach to leverage events for RSC. For the synthesis branch row-wise readout time offsets are injected into attention and fused with RS frame to restore a GS image, while for the warping branch events join the estimation of flow between RS and GS images and obtain another GS image. The two estimated GS images are further combined to give the final GS restoration result. In [1] a transformation is conducted to encode only events between readout time of RS and GS such that row-wise motion can be directly estimated without constant linear motion assumption. It also takes two-branch framework and uses the fusion result as the final prediction. However, the way that they use the events and row-wise readout time for correcting rolling shutter effect , *i.e.*, either directly injecting time offsets to event representation or selecting only events between RS and GS as input, might not be optimal. Involving row-wise readout time in the interaction between RS images and events could help explore more potential of events for the RSC task.

In this work, we present an event-assisted encoder-decoder framework for rolling shutter correction. In the encoding stage, the RS frame and event data are first respectively sent to a separate convolutional layer for shallow feature extraction. They are then separately fed to several transformer layers which are equipped with the proposed time-aware multi-head attention for cross-modal feature enhancement. Since time encodings play different roles in the feature enhancement for each modality, different forms of time encodings are exploited. For the RS image, row-wise time encodings are used because each row in the RS image correspond to the same exposure time; while for event data, channel-wise time encodings

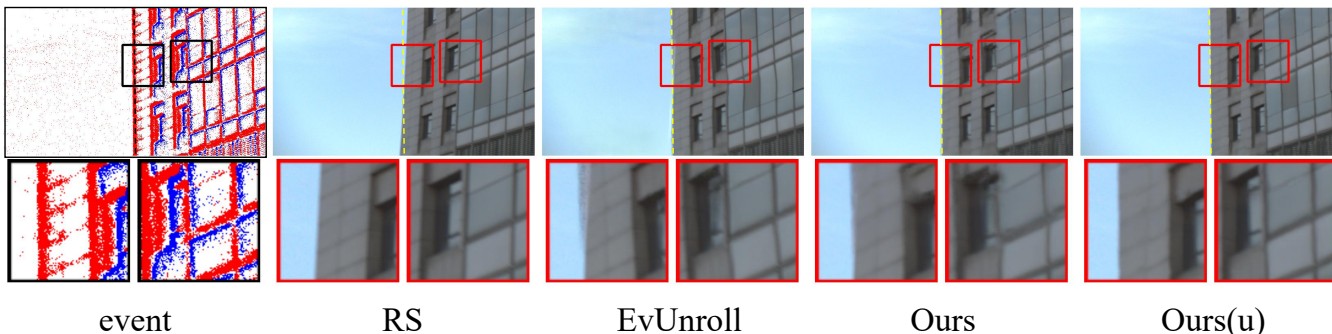

event RS EvUnroll Ours Ours(u)

**Figure 1: An example of rolling shutter correction on real-world data compared with sota method EvUnroll [32]. The yellow dashed line is the GS reference line drawn based on the intermediate event. The label (u) means fine-tuning using self-supervised learning.**

are used due to the use of voxel grid representation. We utilize this characteristic to design time encoding and combine it with multi-head attention to better integrate the information from both modality. In the decoding stage, we produce the corresponding GS image at the desired time stamp. Due to the significant image displacement in the RSC task, directly reconstructing the image from features can be challenging. To address this, we utilize Deformable Cross Attention (DCA) [34] to expand receptive field. By taking both event features and GS time encoding as the queries and RS features as keys, DCA can effectively leverage the motion information with events and appearance information with RS frame to reconstruct a GS image.

Moreover, to improve the model's generalization ability, we designed an self-supervised method with single RS image and event specifically for the RSC task. This method enables us to fine-tune the model on test datasets and real datasets. In summary, our contributions are as follows:

- We proposed a novel architecture specifically designed for the characteristics of the RSC task.
- We proposed a self-supervised loss tailored for the RSC task to enhance the model's generalization and improve its performance on real-world datasets.
- Our method outperforms existing methods in both qualitative and quantitative performance.

## 2 RELATED WORKS

### 2.1 Rgb-based RSC methods

Multiple images methods mostly predict optical flow between two consecutive images according linear motion assumption. Fan *et al.* [2, 4] proposed to calculate optical flow in stage one, and then obtain the final GS image with warped image in stage two, which is similar to the [9] structure. DSUN [15] and SUNet [3] calculate cost volume for multi-scale pyramid features respectively, and distort and fuse the features to restore GS image. JCD [31] simultaneously processes RSC and deblurring tasks by using deformable attention module to fuse bidirectional warped features. Naor *et al.* [17] regarded the RS correction problem as a temporal upsampling problem and proposed additional supervision at the x-t level. Fan *et al.* [5] proposed a single-stage architecture which combines

appearance refinement and undistortion motion estimation for efficient RS correction. Zhong *et al.* [30] adopted a novel setup where two RS images are scanned in different directions (from top to bottom and from bottom to top) simultaneously, extracting symmetrical information for GS recovery. Building upon the same setup, a bidirectional warping module was designed to restore original RS images in [19], followed by self-supervised training based on temporal consistency.

### 2.2 Multi-modality image enhancement

The rich temporal information of events enables them to perform many image enhancement tasks, such as deblurring [10, 14, 18, 20–22, 27, 29], frame interpolation [7, 12, 23, 24, 26], and RSC [1, 32]. REDNet [27] achieves self-supervised deblurring by unfolding a blurry image into multiple sharp images and introducing optical flow constraints between each sharp image. TimeReplayer [7] uses a structure similar to SuperSloMo to interpolate an image from two input images, and one input image and the resulting image can be extrapolated to another input image, achieving self-supervised results. EVDI [29] uses two blurry images taken at different times to restore a sharp image at the same timestamp, achieving self-supervised results. These self-supervised methods mostly rely on either a single blurry image or multiple sharp images.

TimeLens [24] is the first to use both the motion information and the physical properties of event , and it employs two branches for warping and synthesis to perform frame interpolation. EvUnroll [32] uses a similar two-branch structure to handle the RSC problem, reflecting the different time relationships of each row in RS by changing the timemap during warping branch. EvShutter [1] captures the events between RS and GS and estimates the optical flow between RS and GS without using the linear motion assumption.

## 3 METHOD

Here we first briefly introduce preliminary knowledge about event representation and formation of rolling shutter images. Then we will explain how spatio-temporal relationship among events, RS images and GS images could be leveraged in the encoder-decoder framework for event-assisted rolling shutter correction. Finally, the supervision imposed on the model training is presented, which also

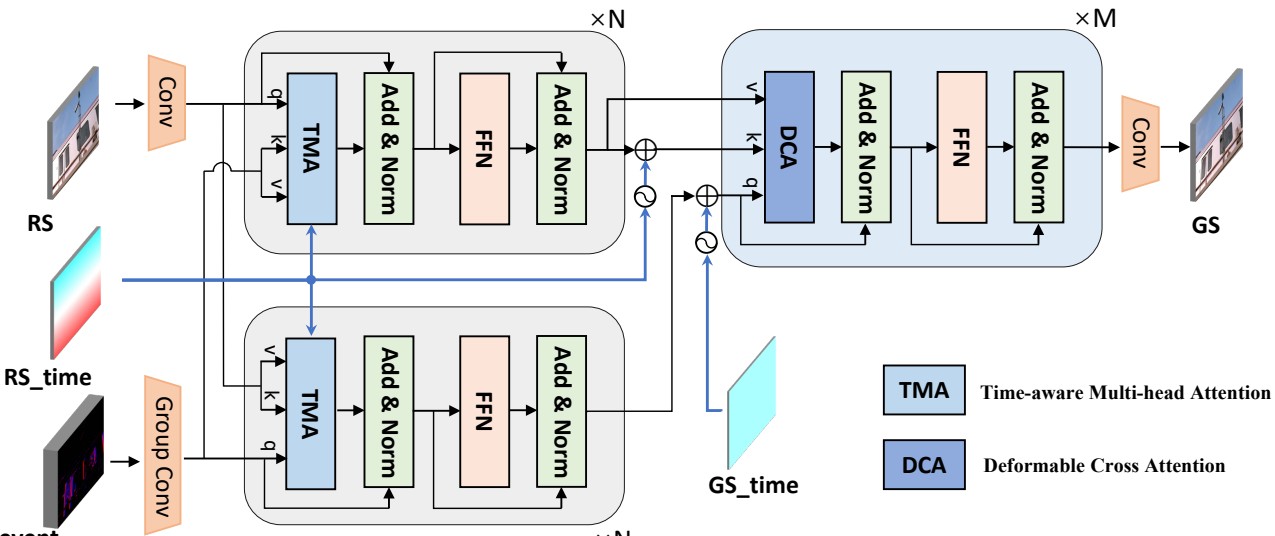

**Figure 2: Overview of our network architecture.**

include both supervised losses and the proposed self-supervised loss.

## 3.1 Preliminary

In an event camera, when the intensity variance of a certain pixel between consecutive time stamps reaches a threshold $C$, an event with a certain polarity would triggered. Typically, a stream of events over a period of time can be represented as:

$$\mathcal{E} = \{e_i\}_{i=1}^N = \{[x_i, y_i, t_i, p_i]\}_{i=1}^N, \tag{1}$$

where $t$ is the triggered time stamp, $(x, y)$ represents the triggered pixel position, and $p$ represents its polarity. A popular way to represent event set is to transform those events into voxel grids [33] as shown below:

$$E(x_l, y_m, t_i) = \sum_{\substack{x_i = x_l \\ y_i = y_m}} p_i \max(0, 1 - |t_i - t_i^*|), \tag{2}$$

where $t_i^\star \triangleq \frac{B-1}{\triangle T}(t_i - t_0)$ represents the normalized time slices. In voxel grids, events are accumulated in adjacent time slices such that each time slice could contain information at different times. Compared with other event representation such as record the quantity and the last moment of the event, voxel grid contains more detailed temporal information.

A rolling shutter image can be treated as a stack of rows of a global shutter image with each one exposed at different time stamps. Even a small object motion or camera motion would cause distortion in the rolling shutter image. Hence a rolling shutter image can be formulated as below:

$$I_{RS}^{(t)} = \sum_{h=1}^H I_{GS}^{(t+(h-M/2)t_r)}[h], \tag{3}$$

where $t_r$ denotes the readout time between consecutive rows; $M$ denotes the number of rows; $t + (h - M/2)t_r$ means the exposure time of the $h$−th row. Since GS are sharp images captured with short exposure time, here the temporal information is represented by the mid-exposure moment rather than the start and end of exposure.

## 3.2 Method

As shown in Fig. 2, the overall model is an encoder-decoder based framework composed of transformers. In encoder, we employ transformers with the proposed Time-Aware Multi-head Attention to model the spatio-temporal relationship among RS images, events and GS images to obtain multimodal features for later GS image decoding. In decoder, we used DCA to extract the appearance information of RS features based on the timestamp of GS, utilizing the motion information of the event features. The network takes a stream of events and an RS image with its exposure parameters as inputs, outputting GS images at any time within the time interval.

**Encoder with Time-aware Multi-head Attention**

Events record almost all continuous brightness change during the exposure time, from the beginning time stamp of the first row of a RS image to the end time stamp of its last row. Such brightness change could be caused by either motion or dynamic lighting. In this task we expect to explore the value of motion information associated with events in the entire exposure period. The encoder starts with a few convolutional layers for shallow feature extraction of the RS frame, while for event data group convolution operation is conducted to retain time ordering information. Simply concatenating representations of two modalities might not be suitable because the features of two modalities are neither spatially aligned nor temporally synchronized. This would cause significant burden to the network in later information fusion and feature enhancement. To fully make use of motion information with event data for RSC, we first resort to powerful transformers to enhance features of both RS frame and events with help of each other. We design a novel Time-aware Multi-head Attention (TMA) for the transformer layer

to enhance each modality. The design principle is to take timing information as a bridge between the modalities of RS frame and event stream.

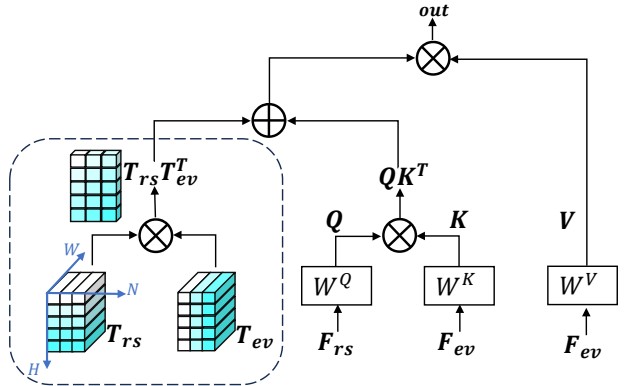

**Figure 3: Time-awareness Multi-head Attention.**

Inspired by the positional encodings in most transformers [25], here we propose to use time encodings in the transformer and leverage them to promote mutual enhancement. As for features of the RS frame $\mathcal{F}_{rs} \in \mathbb{R}^{B \times C \times H \times W}$, its rows correspond to a set of consecutive readout time stamps. Therefore we can directly assign a row-dependent time encodings to RS features. Since each column has the same readout time, we simply denote the time encoding for RS frame as $\mathcal{T}_{rs}^{h,w} = sinusoid(\frac{h}{H}), h \in [0, H), w \in [0, W)$, where $sinusoid$ represents the sinusoidal positional encoding, represented as:

$$sinusoid(p)[2i] = sin(p/10000^{2i/C}),$$
$$sinusoid(p)[2i+1] = cos(p/10000^{2i/C}), \qquad (4)$$

where $i = (0, ..., C/2 - 1)$.

While as for the voxel-grid-like representation of event data $\mathcal{F}_{ev} \in \mathbb{R}^{B \times C \times H \times W}$, groups along channel correspond to consecutive temporal intervals. Therefore, we divide the event features into $N$ groups along channel, with each one corresponding to certain temporal range. Group-channel-wise time encodings are assigned to event features, which can be denoted as $\mathcal{T}_{ev}^{h,w,i} = sinusoid(\frac{i}{N})$ and $\mathcal{T}_{ev}^{h,w} = concat([\mathcal{T}_{ev}^{h,w,i}]), i = (0, ..., N-1)$. To promote interaction between RS features and event features, time encodings of each modality are combined with the corresponding modality features in computation of cross-attention. Inspired by [16], query or key features are concatenated with time encodings and the cross-attention could be computed as below:

$$attention = \text{SoftMax}((QK^T + \mathcal{T}_{rs}\mathcal{T}_{ev}^T)/\sqrt{d})V \qquad (5)$$

However, it is not proper to directly compute the product of $T_{rs}$ and $T_{ev}$ because each time encoding vector for RS frame corresponds to one timestamp while each time encoding vector for event data corresponds to multiple timestamps. Hence we take inspiration from multi-head cross-attention and present time-aware multi-head attention to address this issue. Here we will take the case of RS features as queries and event features as keys and values as an example to explain how Time-aware Multi-head Attention (TMA) works. Since each group of $C/N$ channels in the

time encodings for event features corresponds to a certain temporal range, the dimension of time encodings for RS features are also set to $C/N$. In this way time encodings of two modalities could be easily compared and their similarity could also be measured through product. The $i$−th group event time encodings is $\mathcal{T}_{rs}^{h,w,i} = sinusoid(\frac{i}{N}) \in \mathbb{R}^{B \times C/N \times H \times W}$, and the $i$−th group RS time encodings is $\mathcal{T}_{rs}^{h,w,i} = sinusoid(\frac{h}{H}) \in \mathbb{R}^{B \times C/N \times H \times W}$. In this way, both RS and event features are divided into sub-features similar to the heads in standard multi-head attention, we can compute the product of RS and event time encodings in a time-wise way within each head. Correspondingly, RS features and event features are also divided into $N$ sub-features which are denoted as $\mathcal{F}_{rs}^i$ and $\mathcal{F}_{ev}^i$. Then as shown in Fig 3, time-aware multi-head attention can be represented as:

$$\text{MHA}(Q, K, V) = \text{Concat}(\text{head}^1, ..., \text{head}^N)W^O, \qquad (6)$$

where $\text{head}^i = \text{SoftMax}((Q^i K^{i^T} + \mathcal{T}_{rs}^i \mathcal{T}_{ev}^{i^T})/\sqrt{d})V^i)$, and $Q^i = \mathcal{F}_{rs}W_i^Q, K^i = \mathcal{F}_{ev}W_i^K, V^i = \mathcal{F}_{ev}W_i^V, d = C/N$. As for the case of event features as queries and RS as keys and values, the computation is similar. In summary, by dividing temporal encodings and features for RS frame and event data into multi-heads, TMA able to effectively integrate temporal relationship between the two modalities into the cross-attention for cross-modal enhancement .

### Decoder with Deformable Cross Attention

In the encoding stage, information associated with RS frame and event data is sufficiently explored and fused. The decoder aims at predicting the desired GS image in the middle exposure period or at any specified time stamp based on the enhanced features from encoder. Since there is often significant misalignment between RS and GS frames, we employ deformable cross-attention in the decoder to expand receptive fields and accelerate model convergence. The queries of the deformable cross-attention consist of two components, *i.e.*, enhanced event features from the encoder and the time encoding for desired GS time stamp. The enhanced RS features from the encoder are used as the keys and values of the attention. In this way the desired time and rich motion information provided by event features work as guidance and complement to remove spatial distortion in RS features. Then a GS image is recovered from the spatially corrected RS features with another few convolutional layers.

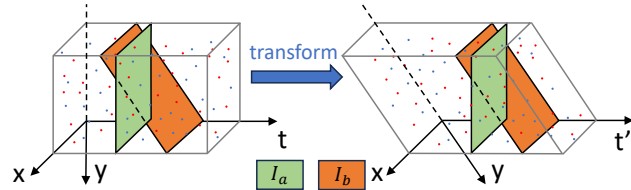

**Figure 4: temporal reference system transform.**

## 3.3 Supervision for Rolling Shutter Correction

The supervision for our RSC framework include both supervised reconstruction loss and a proposed self-supervised loss. We first introduce the proposed self-supervised loss. As shown in Fig. 1,

given a pair of RS and GS images $I_a$ and $I_b$, can we determine which one is RS image and which one is GS image? Obviously, we cannot. Without a prior assumption like "lines must remain straight", we cannot determine if an object appears curved due to rolling shutter effect or if it was originally shaped in that way.

Actually, we could have two different time reference systems based on either GS $t_{gs}^i$ or RS $t_{rs}^i$. In the GS time reference system, each row in GS image $I_a$ has the same exposure time while each row in RS image $I_b$ has different time as shown below:

$$I_a : t_{gs}^{a,1} = t_{gs}^{a,2} = ... = t_{gs}^{a,H}, \qquad (7)$$
$$I_b : t_{gs}^{b,i} = t_{gs}^{b,1} + (i-1) * \delta,$$

where $t_{gs}^{a(b),i}$ represents the $i-$th row's exposure time with respect to GS time reference system, $i \in [1, ..., H]$, $\delta$ denotes the readout time delay between successive rows in $I_b$.

In the RS time reference system, each row in $I_b$ has the same exposure time while the one in $I_a$ has different time as shown below:

$$I_a : t_{rs}^{a,i} = t_{rs}^{a,1} - (i-1) * \delta, \qquad (8)$$
$$I_b : t_{rs}^{b,1} = t_{rs}^{b,2} = ... = t_{rs}^{b,H},$$

where $t_{rs}^{a(b),i}$ represents the $i-$th row's exposure time with respect to GS time reference system. The relation between the two time reference systems can be represented in the transform as shown in Eq. 9 and Fig 4.

$$t_{rs}^i = f(t_{gs}^i) = t_{gs}^i + (i-1) * \delta + b, \qquad (9)$$

where $b$ represents the inception. The choice of time reference system would affect the use of event data because every event is represented with a quadruplet $(t, x, y, p)$ including its time stamp, position, and polarity. By transforming the time value in the event quadruplets, we convert events from one temporal coordinate system to another, which are respectively denoted as $e$ and $e'$ as shown below.

$$e(t_{gs}, x, y, p) = e'(t_{rs}, x, y, p). \qquad (10)$$

Assume we have a pre-trained rolling shutter correction model $f_{RSC}$. By introducing the idea of cycling transform, we could construct a self-supervision loss. Firstly, in the GS time reference system, the RS image $I_a$ with RS and GS timestamp is sent to the RSC model to predict a GS image $I_{o1}$.

$$I_{o1} = f_{RSC}(I_a, e, t_{gs}^a, t_{gs}^b) \qquad (11)$$

By switching to RS time reference system and considering $I_{o1}$ as the input "RS image", we can apply the RSC model again to recover a "GS image" with help of events $\{e\}$.

$$I_{o2} = f_{RSC}(I_{o1}, e', t_{rs}^a, t_{rs}^b) \qquad (12)$$

The recovered "GS image" $I_{o2}$ in RS time reference system should be identical to the original RS image $I_{rs}$ in GS time reference system. Hence the self-supervised loss function could be computed as follows:

$$\mathcal{L}_{self} = ||I_{rs} - I_{o2}||_1. \qquad (13)$$

We provide an x-t plot in the supplementary materials for a better understanding of self-supervised learning.

Finally, our loss function consists of a Charbonnier loss [? ] to reconstruct GT, a perceptual loss [11], and a self-supervised loss.

$$\mathcal{L} = \mathcal{L}_{error} + \alpha \mathcal{L}_{photo} + \beta \mathcal{L}_{self} \qquad (14)$$

## 4 EXPERIMENTS

In Section 4.1, we validate the effectiveness of the design module through ablation experiments. Subsequently, in Sections 4.2 and 4.3, we will compare with existing methods on RSC benchmark datasets. Finally, we will examine the effectiveness of self-supervised effectiveness on different methods and both synthetic and real datasets.

**Table 1: Ablation study**

| Methods | PSNR | SSIM |
|---------|------|------|
| SwinIR [13] | 30.83 | 0.88 |
| w/o TMA | 32.87 | 0.91 |
| w/o DCA | 32.18 | 0.91 |
| ours | **33.53** | **0.92** |

## 4.1 Ablation study

To examine the effectiveness of TMA, we experiment with another simpler time encoding without the design of multi-head attention focusing on events at different time periods. In this approach, RS time encoding is generated based on the exposure time of each row with the same size as the RS feature ($t_{rs}, \mathcal{F}_{rs} \in \mathbb{R}^{B \times C \times H \times W}$), and event time encoding is generated by the mid-exposure time of RS with the same size as the event feature as well ($t_{ev}, \mathcal{F}_{ev} \in \mathbb{R}^{B \times C \times H \times W}$). And their product is add to the product of query and key as Eq. 5.

To validate the importance of receptive field in the RSC task, we conduct a comparative experiment where we replace deformable attention with normal attention. Additionally, we provide a reference experiment with SwinIR to verify if the encoder-decoder structure is better. For SwinIR, we directly concatenate RS, event, and timemap together as the input of the original SwinIR architecture.

As shown in Tab. 1, compared with original SwinIR architecture, our time enconding baesd encoder-decoder structure performance better. This is beacuse without query based decoder, it's hard to build difficult to establish the temporal relationship between RS, GS, and events. Just concatenating all these as inputs can't fully integrate the temporal information of the two modalities. Without deformable attention, there is noticeable decrease in PSNR. This is mainly due to two reasons. First, the network lacks a sufficient receptive field, resulting in failure to recover in scenes with large RS offsets, such as high-speed motion, leading to residual RS artifacts. Additionally, the use of regular cross-attention does not effectively exploit the characteristics of each modality, and the motion information of events is not fully utilized. In addition to its effectiveness, using DCA can also lead to reduced computational complexity and accelerated training. Using a different time encoding instead of TMA will also result in a certain decrease in PSNR. That is because this type of encoding of RS represents only a small time interval at each position while event time encoding represents the entire exposure duration. When performing cross-modality

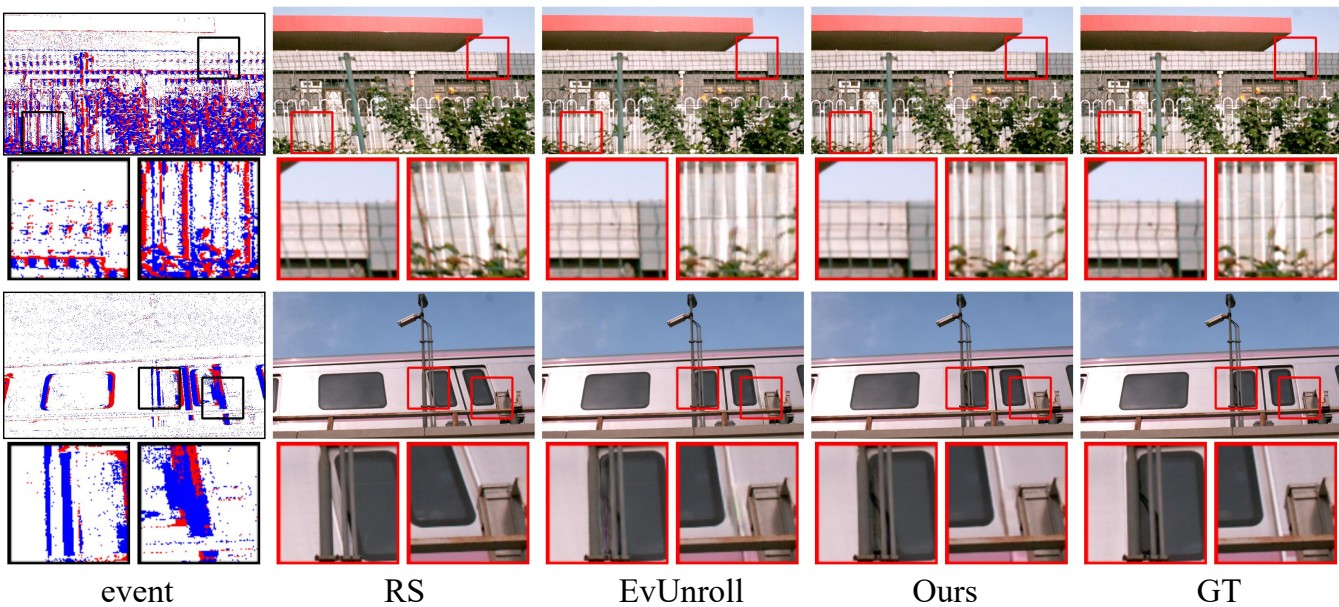

| event | RS | EvUnroll | Ours | GT |

**Figure 5: Visual comparison on Gev-RS dataset. The event frame is generated by the events within the neighborhood of the GS timestamp. The displayed ground truth (GT) in the images corresponds to the mid-exposure time in RS.**

fusion, directly fusing these two can not extract precise temporal information. In scenarios with high requirement on time accuracy, such as high-speed motion, it may lead to local blur in the restored output.

**Table 2: Results on Gev-RS.**

| Methods | events | PSNR | SSIM | LPIPS |
|---|:---:|:---:|:---:|:---:|
| DSUN [15] | ✗ | 23.10 | 0.70 | 0.166 |
| JCD [31] | ✗ | 24.90 | 0.82 | 0.105 |
| IFED [30] | ✗ | 27.16 | 0.85 | 0.097 |
| SelfDRSC [19] | ✗ | 26.58 | 0.84 | 0.113 |
| JAMNet [5] | ✗ | 27.02 | 0.85 | 0.105 |
| EvUnroll [32] | ✓ | 32.16 | 0.91 | 0.061 |
| ours | ✓ | **33.53** | **0.92** | **0.021** |

### 4.2 Experiments on Gev-RS dataset

Gev-RS collects GS videos by using a high-speed camera at 5700 fps. Then GS videos are fed into the event simulator V2E [8] to obtain event data, and generate RS images based on Eq. 3 with 640 × 360 resolution.

In Fig. 5, we compare our method with EvUnroll [32] on Gev-RS dataset. The resulting images in the figure are all GS images at the intermediate exposure time of RS images. In the first case, it can be clearly seen that Evunroll produces artifacts when the foreground and background are close, such as the region between the metal box and the train. Additionally, it also failed to accurately reproduce the reflection of the lamp post in the window.

This is caused by event overlap between the foreground and background. Our approach effectively integrates the two modalities, enabling model to distinguish whether the event is generated by the foreground or background. As a result, our method can avoid the occurrence of similar artifacts. The second example is highly challenging, involving complex scenes with moving objects. Both the small green and white fence in the foreground are difficult targets for restoration. While EvUnroll's recovery of the white fence appears slightly blurred and the green fence exhibits disjointed segments, our method can accurately capture the displacement of small objects to restore the green fence and also recover the white fence without being affected by high-speed motion. Finally, as shown in Tab. 2, we outperform EvUnroll by 1.37 dB in PSNR, 0.040 in terms of LPIPS [28] and far superior to other image based methods.

**Table 3: Results on Fastec-RS.**

| Methods | events | PSNR | SSIM | LPIPS |
|---|:---:|:---:|:---:|:---:|
| DSUN [15] | ✗ | 26.73 | 0.82 | 0.166 |
| JCD [31] | ✗ | 26.48 | 0.82 | 0.105 |
| SUNet [3] | ✗ | 27.06 | 0.83 | – |
| JAMNet [5] | ✗ | 28.70 | 0.87 | 0.093 |
| EvUnroll [32] | ✓ | 31.32 | 0.88 | **0.061** |
| EvShutter [1] | ✓ | 32.41 | **0.91** | **0.061** |
| ours | ✓ | **32.51** | **0.91** | 0.070 |

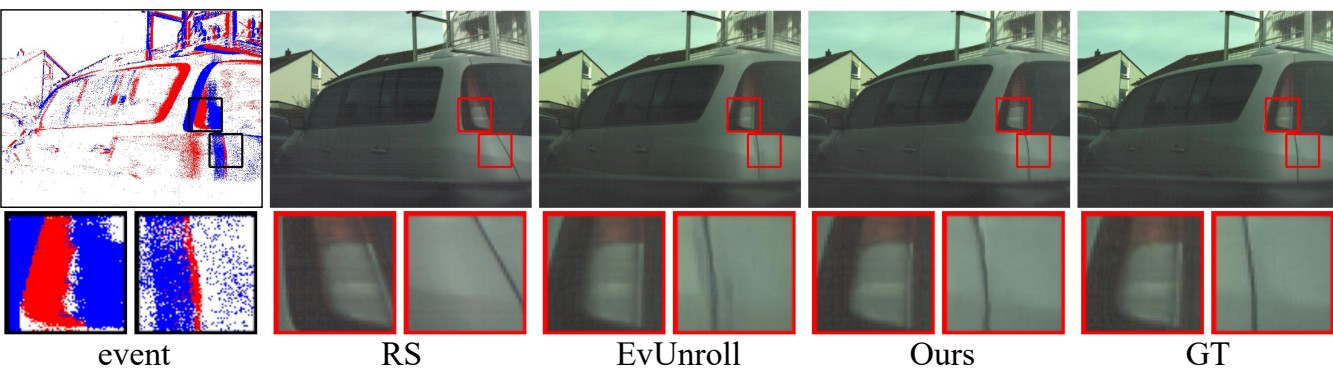

| event | RS | EvUnroll | Ours | GT |

**Figure 6: Visual comparison on Fastec-RS dataset. The event frame is generated by the events within the neighborhood of the GS timestamp. The displayed ground truth (GT) in the images corresponds to the mid-exposure time in RS.**

### 4.3 Experiments on Fastec-RS dataset

Fastec-RS collects GS images at 2400 fps with 640 × 480 resolution, and then simulate event data and RS image in the same way as Gev-RS.

In Fig. 6, We compare our method with EvUnroll [32]. When an object moves too close to the camera, causing significant relative movement, EvUnroll produces noticeable artifacts. Even though evunroll utilizes a dual-branch structure, its warping branch fails under significant displacements, while the synthesis branch lacks detailed handling of temporal information in events. Consequently, when a large number of events occur within a short period due to high-speed motion, the accumulation of unnecessary events leads to blurring. In our method, each head of the TMA focuses on event information at different time frames, enabling us to better address the challenges presented by highly dynamic movements.

As shown in Tab. 3, Our method outperform EvUnroll by 1.16 dB in PSNR but only exhibits only a slight superiority over EvShutter [1]. More results compared with image-based methods on Gev-RS and Fastec-RS will be shown in Supp. Mat..

**Table 4: Results on different methods with our self-supervised learning.**

| Dataset | Methods | PSNR | SSIM |
|---------|-----------|-------|--------|
| Gev-RS | EvUnroll | 32.16 | 0.91 |
|  | EvUnroll(u) | 33.49 | **0.92** |
| Gev-RS | ours | 33.53 | **0.92** |
|  | ours(u) | **33.92** | 0.92 |

### 4.4 Experiments on self-supervison

To validate the effectiveness of self-supervised learning, we conducted experiments on both the test set of a synthetic dataset and a real-world dataset.

As shown in Fig. 7, the left image is the result of training with self-supervised function, while the right image is reversed. With self-supervised learning, the model learned partial data distribution of the test set and removed the artifacts caused by incomplete

restoration, leading to better performance and a 0.49 dB improvement in PSNR. Additionally, we applied the self-supervised loss to EvUnroll, resulting in a 1.33 dB boost, as shown in Tab. 4. This demonstrates the broad applicability of our self-supervised function, which can be applied to different multi-modality methods.

Furthermore, in the Fig. 1 and Fig. 8, we present the results on a real dataset collected by EvUnroll. The yellow dashed line is the GS reference line drawn based on the intermediate event. The initial results with ghosting effects were not good, possibly due to dataset gaps, but after the self-supervised learning, the image quality gradually improved. In the case of building, the boundary the building restored by EvUnroll is quite blurry, as analyzed earlier, primarily due to the limited use of detailed temporal information. Moreover, the restored windows exhibit ghosting and distortion in the glass. Our method outperforms EvUnroll in edge restoration, but it still exhibits ghosting in the windows. Nevertheless, the addition of self-supervised learning results in remarkably clear window restoration. In the next case, self-supervised learning eliminates the direct artifacts between the stick and the base, and prevents the occurrence of puppet distortion in the results of evunroll. It is noteworthy that fine-tuning from the weights pretrained on the synthetic data on real data typically requires just a few hundred iterations.

Both the results on real-world and synthetic datasets demonstrate the effectiveness of the proposed self-supervised loss.

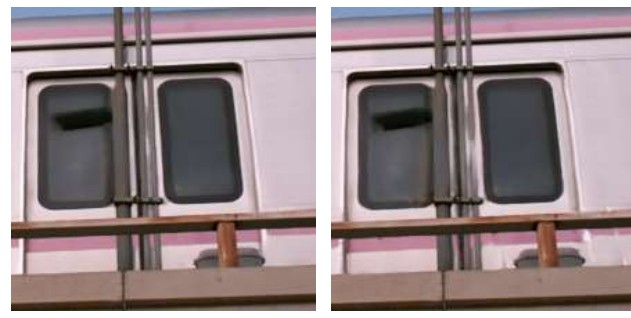

**Figure 7: Visual comparison of self-supervised learning.**

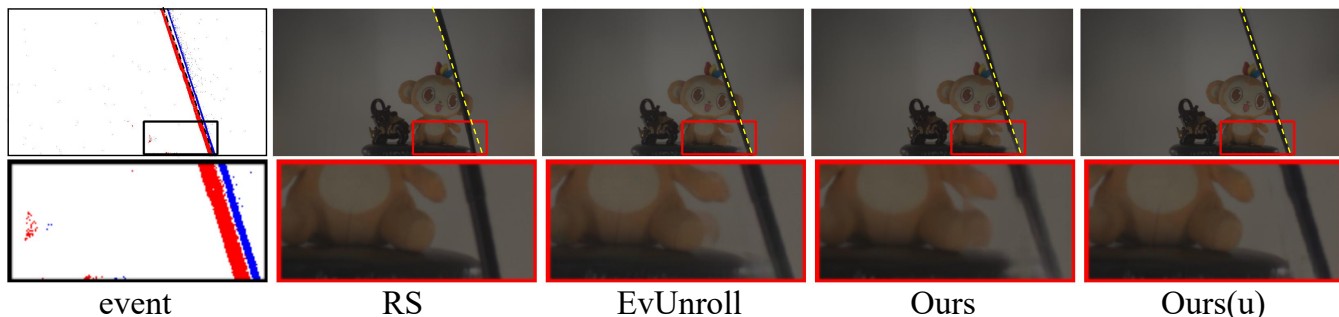

event | RS | EvUnroll | Ours | Ours(u)

**Figure 8: Visual comparison on real-world dataset. The yellow dashed line is the GS reference line drawn based on the intermediate event. The label (u) means fine-tuning using self-supervised learning.**

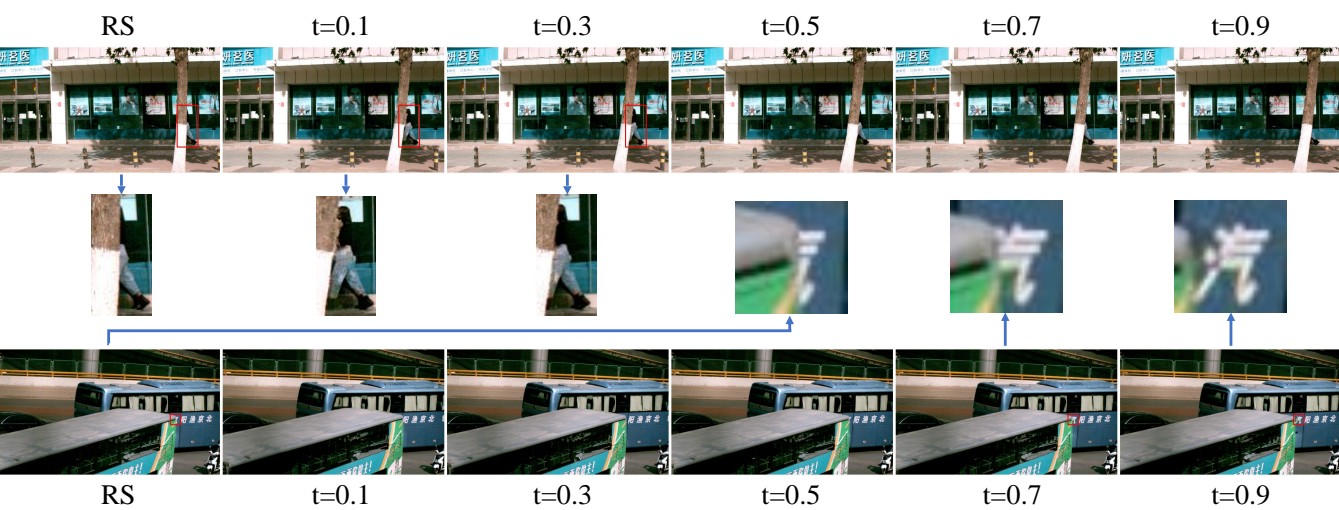

**Figure 9: Visual comparison on arbitrary timestamp.**

## 4.5 Experiments on arbitrary timestamp

To validate the effectiveness of Time-Aware Attention, GS images at different timestamps are generated. As shown in Fig. 9, we present two scenarios, with one featuring a person occluded by trees and the other with text occluded by a bus. Despite occlusions of text and persons by the foreground in the RS images, these areas are effectively recovered.

**Table 5: Performance(PSNR) on different timestamps.**

| timestamp | 0.1 | 0.3 | 0.5 | 0.7 | 0.9 |
|-----------|-------|-------|-------|-------|-------|
| ours | 32.06 | 32.38 | 33.53 | 33.74 | 32.41 |
| EvUnroll | 31.40 | 32.04 | 32.16 | 32.40 | 31.78 |

Additionally, we conduct a quatitative comparison with EvUnroll across different timestamps in Tab. 5. It can be observed that performance decreases as the timestamp $t$ approaches the start or end of the RS exposure time ($t = 0.1, 0.9$). This's because at these moments, the temporal span between RS and GS at the bottom/top of

the image is too large (nearly the whole RS exposure time), leading to poor local reconstruction. In contrast, at the mid-exposure time ($t = 0.5$), the temporal span is relatively smaller (half of the whole RS exposure time). The larger the difference is, the more challenging the RSC task is. As shown in Tab. 5, our method outperforms EvUnroll at all timestamps.

## 5 CONCLUSION

In this work, we propose a encoder-decoder framework for the RSC task. In encoder, we design time encoding as a bridge for the interaction of two modalities, and skillfully integrate it with multi-head attention to make more intricate use of temporal information. In decoder, we use the existing DCA module to integrate motion information of event and appearance information of RS, and employed the GS time encoding as a query for decoding. Additionally, we develop self-supervised loss specific to the RSC task and confirm its effectiveness on both synthetic and real datasets. Our method outperforms existing methods in both qualitative and quantitative performance.

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
