# OpenReview forum: "Event-Guided Rolling Shutter Correction with Time-Aware Cross-Attentions"
_acmmm.org/ACMMM/2024/Conference — MM2024 Poster_

### Official Review · Reviewer_dEG3 · 2024-05-16

**Rating:** 3
**Confidence:** 2

**Summary:**

This paper presents an encoder-decoder framework for the RSC task. In the encoder, a multi-head attention for enhanced temporal information processing is proposed. The decoder utilizes the existing DCA module to merge motion and appearance information. Additionally, a self-supervised loss specific to the RSC task is devised and confirmed effective on both synthetic and real datasets.

**Strengths:**

1. the results on real-world and synthetic datasets demonstrate the effectiveness of the proposed method.
2. The authors have written the paper in an organized manner, making it easy to follow their arguments and understand the presented concepts.

**Limitations:**

1. The proposed Time-awareness Multi-head Attention and self-supervised loss may lack distinctive innovations within the domain of event-based vision, as it appears to amalgamate existing methodologies.

2. Will the checkpoints (ckpts) required for replicating the paper's results be released as open-source?

3. The comparative method only includes EvUnroll. The comparison seems insufficient due to the absence of a comprehensive evaluation against state-of-the-art methods.

**Suitability:**

2

---

### Official Review · Reviewer_VStP · 2024-05-24

**Rating:** 3
**Confidence:** 3

**Summary:**

This paper introduces an advanced encoder-decoder framework that uses high-temporal-resolution event cameras to correct distortions in rolling shutter (RS) images. By incorporating time encoding into cross-attention mechanisms within transformers, the method effectively fuses RS images and event data, leading to superior GS image reconstruction. The approach also includes a self-supervised learning component to enhance generalization. Extensive evaluations on both synthetic and real datasets demonstrate significant improvements over existing methods in terms of PSNR, SSIM, and perceptual quality.

**Strengths:**

1.	Innovative use of event cameras leverages high temporal resolution to capture fine-grained motion details critical for correcting rolling shutter distortions.
2.	Incorporates time encoding in transformers for effective fusion of RS images and event data, improving alignment and accuracy.
3.	Encoder-Decoder Architecture: Utilizes transformers for feature enhancement and deformable cross-attention for handling large displacements, resulting in high-quality GS image reconstruction.
4.	Introduces a self-supervised loss function that improves generalization and performance on new datasets without extensive manual labeling.

**Limitations:**

1. This work involves training with both supervised and self-supervised information. Would it be more reasonable to call it semi-supervised?
2. The abbreviation GS should be defined at its first occurrence (line 22).
3. Minor typo: the star symbol (line 274) is inconsistent with Equation (2).
4. The improvement value should be 0.39dB (Line 771) according to the Table.4.
5. The performance improvement of the self-supervised loss function in Table 4 is not obvious, especially regarding whether SSIM. All of the results are trained under the same random seed?
6. Does the self-supervision loss term work on all datasets? If not, could you please provide examples or references of failures and explain why it doesn't work?

**Suitability:**

3

---

### Official Review · Reviewer_jz7V · 2024-05-24

**Rating:** 5
**Confidence:** 3

**Summary:**

1 This paper presents an event-guided rolling shutter (RS) correction method that corrects image distortion and artifacts caused by rolling shutter through a time-aware cross-attention mechanism.

2 A novel temporal encoding is introduced to better integrate temporal information.

3 A self-supervised loss is proposed for fine-tuning the model on real-world data.

4 The performance is qualitatively and quantitatively satisfactory.

**Strengths:**

1 The proposed temporal encoding is innovative, and the self-supervised loss that extends from it is also interesting.

2 The experiments are comprehensive, thoroughly validating the effectiveness of the method.

3 The manuscript is easy to understand and well-structured.

**Limitations:**

1 There are some formatting issues, such as the text in Figure 4 being displayed incorrectly in some PDF readers. L523 lacks a reference citation.

2 The term "Deformable" typically refers specifically to Deformable Convolution Network. It may be misleading in this context of the paper.

3 In the Supp, unsupervised training is mentioned. Does this mean the training is conducted solely with self-supervised loss? If so, it would be best to provide a qualitative and quantitative analysis of the final training results. In fact, the self-supervised loss can serve three purposes: joint training with supervised loss, fine-tuning on the test set, and unsupervised training. It would be best to clarify the usage of the self-supervised loss in the experiments.

4 In some tables, the best results are highlighted in bold, while others are not. There should be consistency in these.

**Suitability:**

3

---

### Official Review · Reviewer_cEtH · 2024-05-25

**Rating:** 3
**Confidence:** 3

**Summary:**

This paper proposes an event-guided rolling shutter correction framework using an encoder-decoder transformer. In the encoder, the authors incorporate time encoding to connect the two modalities, cleverly integrating it with multi-head attention to better utilize temporal information. For the decoder, the authors use the DCA module to merge the motion data from events with the appearance data from RS, and apply GS time encoding as the query for decoding. Besides, a self-supervised loss for the rolling shutter correction task is designed to improve the performance and enhance the generalization in real-world scenarios. The results show that the proposed method outperforms all competitors in two simulated datasets (i.e., Gev-RS and Fastec-RS).

**Strengths:**

1. As far as I know, this work is an early attempt to address the rolling shutter correction problem using an event camera, and it indeed has potential applications in real mobile devices.
2. The authors provide comprehensive experiments that demonstrate the capability to resolve undesired distortions and artifacts caused by rolling shutter effects during fast motion.

**Limitations:**

1. Although event cameras have asynchronous and high temporal resolution characteristics, excessively long time windows in high-speed scenarios can still lead to motion blur. Therefore, determining the appropriate length of the time window for different scenarios is crucial. In real-world applications, finding a reasonable time window may not always be feasible. The authors do not provide an analysis of the impact of different time windows on rolling shutter correction results.
2. In high-speed scenarios, the resulting motion blur might further complicate the issue, potentially making it difficult to distinguish between distortions caused by motion blur and those caused by rolling shutter artifacts. It might be more meaningful to study this problem in conjunction with deblurring tasks.
3. The authors chose PSNR and SSIM as metrics, which may not be effective in assessing distortions caused by rolling shutter effects. Other factors, such as background influences, might have a greater impact on these metrics than the rolling shutter distortions themselves. Although the authors present some effective visual results, it is crucial to establish a quantitative metric specifically for rolling shutter distortion.
4. The authors propose a self-supervised loss function as an innovation, but the various weight parameters within the loss function could significantly impact the results. The authors did not provide ablation studies to explore these effects.
5. The paper has some areas for improvement in writing, such as the usage of definite and indefinite articles like "a" and "an" in the conclusion.

**Suitability:**

2

---

### Meta-Review · Area_Chair_aZFk · 2024-07-01

**Recommendation:** Accept (Poster)
**Confidence:** 4

**Metareview:**

Based on the summaries and final ratings provided by the reviewers, it is clear that while there are some concerns, the paper also has significant strengths and innovative aspects, and the author's rebuttal also effectively addressed these concerns. The decision to accept this paper is supported by the following considerations:

1. The paper presents an innovative approach to rolling shutter correction using an event-guided framework with a novel temporal encoding method, which is a valuable contribution to the field.

2. The self-supervised loss function is a creative addition that enhances the model's performance and generalization capabilities.

3. The authors have addressed the concerns raised by the reviewers in their rebuttal, which indicates their willingness to improve the work further.

Despite the majority of the reviewers giving negative scores, I did not find compelling reasons in their comments to reject this article. Moreover, the reviewers requested the authors to supplement many experiments. Although the authors are not obliged to provide additional experimental results, their rebuttal effectively addressed these concerns. Therefore, I still lean towards accepting this manuscript.